# Extracellular Albumin Covalently Sequesters Selenocompounds and Determines Cytotoxicity

**DOI:** 10.3390/ijms20194734

**Published:** 2019-09-24

**Authors:** Wenyi Zheng, Roberto Boada, Rui He, Tingting Xiao, Fei Ye, Laura Simonelli, Manuel Valiente, Ying Zhao, Moustapha Hassan

**Affiliations:** 1Department of Laboratory Medicine, Karolinska Institute, 141 86 Huddinge, Sweden; rui.he@ki.se (R.H.); Ying.Zhao.1@ki.se (Y.Z.); 2Centre GTS, Department of Chemistry, Autonomous University of Barcelona, 08193 Barcelona, Spain; roberto.boada@uab.cat (R.B.); Tingting.Xiao@uab.cat (T.X.); manuel.valiente@uab.cat (M.V.); 3Division of Functional Nanomaterials, Royal Institute of Technology, 100 40 Stockholm, Sweden; feiy@kth.se; 4CELLS-ALBA Synchrotron Radiation Facility, Carrer de la Llum 2-26, 08290 Barcelona, Spain; lsimonelli@cells.es; 5ECM, Clinical Research Center, Karolinska University Hospital, 141 86 Huddinge, Sweden

**Keywords:** selenium, albumin, cytotoxicity, cellular uptake, X-ray absorption spectroscopy

## Abstract

Selenocompounds (SeCs) are well-known nutrients and promising candidates for cancer therapy; however, treatment efficacy is very heterogeneous and the mechanism of action is not fully understood. Several SeCs have been reported to have albumin-binding ability, which is an important factor in determining the treatment efficacy of drugs. In the present investigation, we hypothesized that extracellular albumin might orchestrate SeCs efficacy. Four SeCs representing distinct categories were selected to investigate their cytotoxicity, cellular uptake, and species transformation. Concomitant treatment of albumin greatly decreased cytotoxicity and cellular uptake of SeCs. Using both X-ray absorption spectroscopy and hyphenated mass spectrometry, we confirmed the formation of macromolecular conjugates between SeCs and albumin. Although the conjugate was still internalized, possibly via albumin scavenger receptors expressed on the cell surface, the uptake was strongly inhibited by excess albumin. In summary, the present investigation established the importance of extracellular albumin binding in determining SeCs cytotoxicity. Due to the fact that albumin content is higher in humans and animals than in cell cultures, and varies among many patient categories, our results are believed to have high translational impact and clinical implications.

## 1. Introduction

Selenocompounds (SeCs) refer to a broad category of selenium (Se)-containing chemicals. SeCs typically feature strong redox activity and could generate abundant reactive oxygen species (ROS) leading to induction of cell death, due to the Se element [1]. This trait is being extensively translated into cancer chemotherapy [2]. It is noteworthy that some types of cancer have been reported to be more susceptible to oxidative stress than normal cells [3,4]; SeCs hold the promise to outperform conventional chemotherapeutics in achieving targeted therapy [5]. In addition, upregulation of antioxidant defense system has been shown to underlie the development of chemoresistance [6]; thus, concomitant treatment of SeCs has been reported to re-sensitize resistant cancer cells [7]. In fact, two SeCs, selenite [8] and ethaselen [7], have been already used in clinical trials. Inspired by the promising effects of SeCs and in order to further improve their treatment efficacy, several researchers have started to introduce Se into current treatments, including both small molecules (i.e. aspirin and zidovudine) [9,10] and antibodies (trastuzumab and bevacizumab) [11].

Most of the SeCs investigated in cancer chemotherapy are synthetic, with a few of natural origin, like selenite and Se dioxide. Many synthetic SeCs derive from precursor compounds and could be categorized by functional moieties, like selenourea, diselenide, selenoester, selenocyanate, seleninic acid, selenazolone, and others [12]. The cytotoxicity of different SeCs is rather diverse, and a clear structure–activity correlation has not been established so far. It is generally accepted that methyl selenol and hydrogen selenide are the pharmacologically active metabolites. These metabolites bind covalently to cellular components and/or react with molecular oxygen to generate ROS in the redox cycles [13]. The high variability of SeCs cytotoxicity against the same type of cancer cells has been largely attributed to their differing potentials to be metabolized into methyl selenol and hydrogen selenide [14,15,16]. This mechanism has not yet been explicitly evidenced due to the poor understanding of the entire metabolic pathway from one side. On the other side, most of the available results are based on detection of the total Se element due to the lack of speciation methods. Moreover, the cytotoxic effect of one SeC can differ several-fold between different cancer cells [17]. Even though a simple explanation of different cancer cell features, like the strength of the antioxidant defense system [18] and expression level of effector proteins [19,20,21], is plausible, the overall picture is still missing. 

The available understanding of SeC cytotoxicity is very limited to intracellular factors. In comparison, scattered information suggests that several SeCs show strong albumin binding; thus, we hypothesized that cellular uptake and downstream cytotoxicity might be orchestrated by extracellular albumin. To address this issue, we selected four SeCs of different structural moieties and studied the effect of extracellular albumin on their cellular uptake and cytotoxicity in vitro.

## 2. Results

### 2.1. Concomitant Treatment with Albumin Decreased Selenocompound Cytotoxicity

Four anti-cancer SeC candidates with different structural moieties were included in this study (Figure 1a). The selenocyanate compound p-XSC (p-xyleneselenocyanate) was shown to be highly toxic to the murine acute myeloid leukemia cell line C1498 in a concentration-dependent manner, while its cytotoxicity was completely abrogated after addition of 1% BSA (bovine serum albumin) in the complete medium (Figure 1b). The cytotoxicity of p-XSC to other two human cell lines, HL60 (acute promyelocytic leukemia) and HUVEC (human umbilical vein endothelial cell), was decreased in similar way to the results with C1498 cells (Figure 1c,d). In order to investigate whether this pattern applied to other categories of SeCs, e.g., selenazolone, seleninic acid, and diselenide, we examined the cytotoxicity of ebselen, MeSeA (methylseleninic acid), and CysSe_2_ (selenocystine) on C1498 cells with or without 1% BSA (Figure 1e–g). In a similar manner, their cytotoxicity was substantially compromised in the presence of 1% BSA.

Since the complete medium was routinely supplemented with 10% (for C1498 and HL60) or 5% (for HUVEC) of serum in this study, albumin was already present at a final concentration of approximately 0.2% or 0.1%, respectively. In order to exclude the effect of serum-derived albumin, we examined the cytotoxicity of SeCs on C1498 cells cultured in FBS-free medium (Appendix A). A short treatment period of 2 h followed by an instant highly sensitive bioluminescence assay was used to minimize cell death due to nutrient deprivation. Again, addition of albumin did protect the cells from cytotoxicity of p-XSC and ebselen (Appendix A), while MeSeA and CysSe_2_ failed to induce significant cell death at the chosen concentration and treatment duration and the protective effect of albumin was not observed (Appendix A).

### 2.2. Reduced Cellular Uptake of Selenocompounds Underlined the Diminished Cytotoxicity

Next we aimed to investigate the mechanism whereby albumin decreases SeC efficacy. Although stand-alone albumin had no effect on cell proliferation (Appendix A), it is well recognized to have strong ligand-binding ability, and might orchestrate subsequent cellular uptake of SeC (Figure 2a) [22]. Following this concept, we measured intracellular Se amount upon addition of albumin. After treating C1498 cells with SeCs for 30 min, no cell death was observed using trypan blue assay (data not shown), thereby dismissing mutual free transport of Se across the cell membrane. Intracellular Se element was obviously increased upon treatment with p-XSC or ebselen for 30 min compared to basal levels (Figure 2b), but this was not observed in the case of MeSeA or CysSe_2_ (Appendix A). Co-treatment of 1% BSA with p-XSC or ebselen massively decreased the intracellular Se content down to approximately the basal level. On the other hand, preclusion of serum-derived albumin through deprivation of FBS significantly increased uptake of all SeC. Again, supplementation of BSA into FBS-free medium reduced cellular uptake remarkably (Figure 2b and Appendix A). In general, an inverse relationship between extracellular albumin content and intracellular Se level was observed (Appendix A).

### 2.3. Selenocompounds Covalently Bind to Extracellular Albumin

X-ray absorption spectroscopy (XAS) is an element-specific spectroscopic tool that can provide information about the coordination environment and chemical state of the element; it was therefore chosen to study the species transformation of Se. To elucidate the binding between SeC and albumin, the putative conjugate between SeC and BSA (SeC–BSA) was purified. The species transformation in the context of cell culture medium was also investigated at the concentration where SeC displayed notable cytotoxicity to C1498 cells (based on Figure 1b). SeC was mixed with FBS (the sample was abbreviated as SeC–FBS) rather than complete medium to avoid sample dilution and ease lyophilization, and the SeC/FBS ratio was kept at the same level as in the complete medium (Appendix A). For instance, if SeC is toxic at a certain concentration in complete DMEM (containing 10% FBS), its concentration in FBS should be 10 times higher.

Both the X-ray absorption near edge spectra (XANES) and extended X-ray absorption fine structure (EXAFS) of SeC–BSA markedly differed from those of the corresponding SeC (Figure 3a,b), indicating a change in the coordination environment of Se. It is noteworthy that the EXAFS signals were similar among all SeC–BSA conjugates (Figure 3b). The pseudo-radial distribution functions obtained after Fourier transforming of the EXAFS were dominated by the first coordination shell contribution, which was similar for all SeC–BSA conjugates (Figure 3c). The EXAFS modeling corroborated the Se–S binding in a similar fashion (Appendix A). Further modeling of the pseudo-radial distribution function of SeC–FBS also supported the idea that Se coordinates with S (Table 1). In the case of CysSe_2_ and ebselen, the EXAFS fitting indicated that 40% of SeC added into FBS transformed into SeC–BSA, while all p-XSC and MeSeA added into FBS transformed into SeC–BSA. Surprisingly, ebselen also showed a high proportion of a Se metabolite with a Se–Se bond. 

### 2.4. The Role of Albumin and Small Molecular Thiols in Selenocompound Transformation

Small-molecule thiols like glutathione (GSH) and cysteine (Cys) are present alongside albumin in fresh and conditioned cell culture medium. These thiols have been suggested to react and form small-molecule conjugates with SeC (SeC–SM) [23]. In order to clarify the implication of SeC–SM, we chose p-XSC, which has shown high reactivity with GSH and Cys (the conjugate was abbreviated as p-XSC-SM). First of all, we used liquid chromatography–mass spectrometry (LC-MS) to quantify free p-XSC in complete medium with or without 1% BSA. As shown in Figure 4a, free p-XSC was undetectable when its concentration was less than 12 µM, while only 0.62% and 3.24% of the total amount was detected when the concentration reached 16 and 20 µM, respectively. Moreover, free p-XSC was no longer detectable in complete medium supplemented with 1% BSA. 

To investigate the role of p-XSC-SM, p-XSC was prepared in complete medium at a concentration where no free p-XSC was present. The solution was further filtered to remove p-XSC-BSA (the conjugate between p-XSC and BSA) using a centrifugation device with a molecular weight cut-off of 10 kDa (Figure 4b). The filtrate was then analyzed for the presence of p-XSC-SM, cellular uptake, and toxicity profiles. Using a sensitive method previously developed by our group [24], p-XSC-SM was not detected in either fresh or C1498-conditioned complete medium (Appendix A). In parallel, cellular uptake (Figure 4c) and cytotoxicity (Figure 4d) were not observed for the filtrate.

Besides the removal of SeC–BSA in complete medium, we also examined whether extra GSH or Cys could compete for SeC with BSA and thus improve the cytotoxicity. Actually, concomitant treatment with GSH or Cys did not improve the cytotoxicity; on the contrary, p-XSC cytotoxicity was further decreased (Figure 4e and Appendix A). A similar pattern was observed when we examined ebselen, MeSeA, and CysSe_2_ (Appendix A). 

### 2.5. Albumin was Actively Endocytosed

As each BSA molecule contains in theory only one free thiol available for SeC binding, the SeC–BSA conjugate was expected to largely resemble pristine BSA in terms of cellular transport. In light of the mechanism whereby SeC–BSA formed in extracellular compartments enters the target cell, we analyzed cellular fluorescence intensity following FITC–BSA (fluorescein-isothiocyanate-labeled BSA) treatment. In fact, a fluorescence signal was clearly detected microscopically in C1498 cells (Figure 5a), as well as by using a plate reader (Figure 5b), in a concentration-dependent manner. More importantly, co-treatment with BSA significantly decreased the fluorescence signal (Figure 5c), suggesting a competitive uptake mechanism.

## 3. Discussion

SeCs are well known nutrients, and promising drug candidates for both cancer and neurodegenerative disorders. However, tremendous variability in cytotoxicity, treatment efficacy and side effects has been reported. Available data have shown several confounders underlying the described variability, like effector protein expression level, secretion of thiols, and strength of antioxidant defense system. In contrast, our present investigation was the first study to reveal an antagonistic effect of extracellular albumin on SeC cytotoxicity, which was applicable for at least four major categories of SeCs, namely selenocyanate, diselenide, seleninic acid, and selenazolone.

The role of albumin in cell proliferation has been widely discussed, with inconclusive results [25,26]. In the present investigation, our results showed a slight suppressive effect of albumin on C1498 cell proliferation (Appendix A). These results exclude the likelihood that an albumin-derived proliferative effect antagonized SeC cytotoxicity. Furthermore, albumin has been widely reported to have antioxidative action that is executed through free-radical-trapping and ligand-binding [22]. In contrast, SeC cytotoxicity is associated with the generation of intracellular reactive oxygen species and induction of oxidative stress. Given the ubiquitous presence of the antioxidant defense system (e.g., glutathione and thioredoxin pathways) as well as other cellular components (e.g., lipids and proteins) in cells, we assumed that the strong antagonistic effect of albumin on SeC cytotoxicity is unlikely due to the trapping of free radical intracellularly. In other words, our hypothesis was that albumin binds SeC extracellularly and downregulates subsequent cellular uptake and toxicity (Figure 2a). Consistently, measurements of intracellular Se demonstrated an inverse relationship between extracellular albumin content and cellular uptake of SeC (Appendix A).

SeCs have been sparsely reported to bind albumin. In the present study, we further utilized the robustness of XAS to mechanistically dissect the binding profiles. XAS requires minimal sample preparation prior to data acquisition, and could thus preserve original species composition and allow for speciation in a complex biological matrix. Results from the XAS analysis substantiated a covalent Se-S binding model and also validated strong albumin binding in biologically relevant contexts (Figure 3).

Besides albumin, there exist other thiols like GSH and Cys that might participate in SeC transformation [23]. However, our results indicated the absence of small-molecule conjugates in either fresh or conditioned cell culture medium (Appendix A). Moreover, the addition of extra GSH or Cys decreased SeC cytotoxicity as a result of their intrinsic antioxidative functions, rather than increased cytotoxicity through shifting the formation of SeC–BSA towards SeC–SM (Figure 4). These findings collectively highlighted a predominant role of albumin in SeC transformation, in contrast to a negligible role of small-molecule thiols. The importance of albumin might derive from the high reactivity of Cys_34_ residue [22] as well as high concentrations of albumin in common biological matrices. A previous study reported that extracellular Cys could increase the uptake and cytotoxicity of selenite [27]. This is seemingly in contrast with our finding that albumin, rather than small-molecule thiols, is the main determining factor for cellular uptake as well as toxicity. The disagreement might be due to two factors. The first is based on the stepwise species transformation of selenite: (1) reduction catalyzed by GSH or Cys into more reduced forms (e.g., selenodiglutathione); (2) albumin binding. Therefore, the facilitator role of Cys in selenite uptake probably originates from its action on the former transformation step. The second important factor is that the authors used one single concentration of albumin.

Our results clearly marked an antagonistic effect of albumin on SeC cytotoxicity. For treatments/applications that rely on the induction of oxidative stress, including cancer therapy, antibiotics, and antiviral treatment [28], we believe that albumin content should be considered. It is noteworthy that the albumin content in most cell culture media is around 0.2% (range 0.1–0.4%), while that of laboratory animals is 3% and of human plasma is 5% (range 3.5–5.5%). This variability might account for the widely reported heterogeneity in cytotoxicity, and alert researchers to potential discrepancies in translational studies. Furthermore, albumin levels might fluctuate dramatically depending on the pathophysiological background underlying the disease, which could in turn determine the treatment efficacy and adverse effects, including drug-related toxicity of SeCs. Preclusion of albumin binding prior to entrance of target cells could probably improve SeC efficacy, like encapsulation of SeCs into nanomaterials and chemical masking of the functional moiety. This notion necessitates further exploration in vitro and in vivo. Since co-treatment of albumin has been found to decrease SeC uptake, we assume that other biological and/or pharmacological functions of SeCs besides cytotoxicity might be similarly antagonized. 

The diminished cytotoxicity of SeCs in the presence of albumin might have been because the macromolecular SeC–BSA conjugate enters cells much more clumsily than free SeCs. In contrast, our data on the speciation and cytotoxicity of p-XSC indicate that p-XSC-BSA was well internalized and highly cytotoxic (Figure 4a and Appendix A). There is also growing evidence about the systemic toxicity of SeCs in animals and human, where blood albumin content is even higher than the concentration examined in the present study (2.2%–5% vs. 1.2%), emphasizing that SeC–BSA could be taken up. The uptake mechanism of SeC–BSA is not fully understood, however, the competitive inhibition phenomenon might indicate the involvement of albumin scavenger receptors (Figure 5) [29,30]. Early studies had found tumor-specific distribution of Se and utilized selenite as radiotracer for tumor detection [31]. Our study might provide a mechanistic perspective that formation of SeC–albumin conjugates in vivo directs SeCs to the tumors with high demand for albumin. In this sense, exploitation of the albumin scavenger receptors axis might impact SeC uptake and cellular response.

Even though both free SeCs and SeC–BSA could be internalized, it is unclear whether the internalization efficiency of free SeC is superior to that of the corresponding SeC–BSA, as is the relative contribution of each form in the overall cellular uptake. This limitation of our study was due to the fact that the SeC–BSA samples probably contained excess BSA that could competitively inhibit SeC–BSA uptake. When comparing the overall uptake of the four SeCs examined herein, the most efficient uptake was shown to be for p-XSC, followed by ebselen and then MeSeA/CysSe_2_ (Figure 2b and Appendix A). These compounds have varied albumin binding degrees, and thus existed in different ratios of free vs. albumin-bound form. Although the internalization efficiency of free SeCs could be correlated with the hydrophobicity thereof, that of albumin-bound forms could not be compared. To address this question, it would be helpful to obtain different SeC–BSA with the same Se level per molecule, as well as absolute absence of BSA.

## 4. Materials and Methods

### 4.1. Chemicals

MeSeA (Catalog No. 541281), CysSe_2_ (Catalog No. 545996), GSH (Catalog No. G6013), N-methylmaleimide (Catalog No. 389412), tris(2-carboxyethyl)phosphine (Catalog No. 75259), and bovine serum albumin (BSA; Catalog No. A7030) were purchased from Sigma-Aldrich (St. Louis, MO, USA). Ebselen (Catalog No. ALX-270-097) and p-XSC (Catalog No. ab142600) were bought from Enzo Life Sciences (Farmingdale, NY, USA) and Abcam (Cambridge, UK), respectively. Cys (Catalog No. 1.02452.0025) was bought from Merck-Millipore (Burlington, VT, USA).

### 4.2. Cell Culture

C1498, HL60, and HUVEC cells were bought from the American Type Culture Collection (Manassas, VA, USA). C1498 cells were cultured in DMEM (ThermoFisher Scientific, Waltham, USA; Catalog No. 41966) supplemented with fetal bovine serum (FBS; 10%; ThermoFisher Scientific; Catalog No. 41966) and penicillin–streptomycin (1X; Sigma-Aldrich; P4333). HL60 cells were cultured in RPMI1640 (ThermoFisher Scientific; Catalog No. 41966) supplemented with 10% FBS and 1X penicillin–streptomycin. HUVEC cells were grown in endothelial cell media MV2 (PromoCell, Heidelberg, Germany; Catalog No. C22022). The medium with appropriate supplements is herein generally referred as complete medium. All cells were maintained at 37 °C with 5% CO_2_, and routinely assayed by trypan blue. Only cells under their exponential growth phase and with ≤ 5% trypan blue positivity were used for experiments.

### 4.3. Preparation of Selenocompound Stock Solutions

SeC stock solutions were all of 1 mg/mL. Ebselen and p-XSC were prepared in dimethyl sulfoxide. MeSeA and CysSe_2_ were dissolved in H_2_O, and the pH thereof was adjusted to around 7. SeCs were further diluted to the desired concentrations with appropriate medium.

### 4.4. Assessment of Cell Viability

C1498 or HL60 cells were seeded in 96-well plates (1 × 10^4^ cells/well) and immediately treated with SeC; 6 × 10^3^ HUVEC cells were seeded per well and left to attach for 24 h prior to SeC treatment. Following treatments of different durations, cell viability was determined by either WST-1 (Sigma-Aldrich; Catalog No. 11644807001) or CellTiter-Glo (Promega, Madison, WI, USA; Catalog No. G7571) kit. Cells treated with SeC solvent were used as the control. The well containing SeC of the highest concentration and WST-1 or CellTiter-Glo reagent was used for background calibration.

### 4.5. Measurement of Intracellular Selenium Element

A total of 8 × 10^6^ C1498 cells were suspended in 4 mL medium containing SeC. After incubation for 30 min, cells were rinsed with PBS for three times, lysed with 0.4 mL buffer (Promega, Catalog No. E153A), and centrifuged (1 × 10^4^ g, 10 min). The afforded supernatant was diluted with H_2_O to 3 mL and measured by inductively coupled plasma atomic emission spectroscopy (iCAP 6500, ThermoFisher Scientific).

### 4.6. Preparation of Albumin Conjugate and Serum Mixture of Selenocompounds

SeC was mixed with BSA or FBS according to the formulations listed in Appendix A and kept at room temperature for 30 min. The crude material of SeC–BSA was processed by size exclusion chromatography (Superdex 200 Increase 10/300 GL column from GE Healthcare; mobile phase of 5% acetonitrile at a flow rate of 0.75 mL/min) to remove free SeC, and the eluate fractions corresponding to BSA were pooled and lyophilized. SeC–FBS was directly lyophilized.

### 4.7. Selenium Speciation by X-Ray Absorbance Spectroscopy

The Se K-edge XAS experiment was conducted at CLAESS beamline in ALBA synchrotron (Barcelona, Spain) [32]. The measurement was performed in QEXAFS mode using a Si(311) double-crystal monochromator. Around 20 mg lyophilized SeC–BSA or SeC–FBS was pressed into 5 mm diameter pellets, and the data were acquired in fluorescence mode using a five element Silicon drift detector. For the reference compounds, the appropriate amount of SeC (calculated based on an optimum absorption jump in transmission mode) was well mixed with cellulose prior to pellet preparation. Transmission was measured with ionization chambers filled with appropriate mixtures of N_2_ and Kr gases. All measurements were performed at liquid N_2_ temperature to minimize radiation damage. Data analysis was performed with Demeter software package (version 0.9.26) [33].

### 4.8. Speciation of p-Xyleneselenocyanate by Liquid Chromatography–Mass Spectrometry

p-XSC was prepared in DMEM, complete DMEM, or complete DMEM supplemented with 1% BSA. To extract free p-XSC, 50 µL acetonitrile was added to 20 µL SeC-containing matrices, followed by vigorous vortex. The mixture was further centrifuged (3 × 10^4^ g, 10 min) and the afforded supernatant was collected. To extract total p-XSC, 10 µL N-methylmaleimide (0.2 mmol/L in dimethylacetamide) and 5 µL tris(2-carboxyethyl)phosphine (0.1 mmol/L in H_2_O) were successively added into 10 µL SeC-containing matrices. The mixture was left at room temperature for 10 min. A measure of 50 µL acetonitrile was then added, followed by vigorous vortex and centrifugation. A measure of 2 µL supernatant was injected for LC-MS (TSQ Quantum Ultra, ThermoFisher Scientific) analysis, using the parameters listed in Appendix A.

### 4.9. Assessment of Cellular Uptake of Albumin

C1498 cells were cultured in phenol red-free complete DMEM (ThermoFisher Scientific; Catalog No. 21063029). C1498 cells were treated FITC–BSA (Sigma-Aldrich; Cat. No. A9771) for 14 h and rinsed three times with PBS prior to assessment of cellular fluorescence intensity by plate reader or fluorescence microscope. For the competitive inhibition experiment, cells were treated with FITC–BSA (500 µg/mL), BSA (500 µg/mL) or FITC–BSA plus BSA (500 µg/mL each). Fluorescence intensity in untreated cells was defined as the basal level. The control group referred to the cells that were briefly mixed with FITC–BSA (500 µg/mL) and immediately rinsed.

### 4.10. Statistics

Data analysis was performed on Microsoft Office Excel and GraphPad Prism, and is shown as mean ± standard deviation unless otherwise described. Two-sided Mann–Whitney test was applied to compare means between groups. *p* ≤ 0.05 was considered statistically significant and is marked as *.

## 5. Conclusions

Substantial selenium-based treatments are currently being investigated for cancer therapy, with different efficacies and heterogeneous clinical outcome reported. The mechanism behind the inconsistent treatment efficacy has until now been exclusively ascribed to intracellular events (e.g., formation of active metabolites, strength of antioxidant defense system, and expression level of effector proteins). This study, however, highlighted extracellular albumin binding as a predominant factor in determining the cytotoxicity of selenocompounds. Since albumin content in humans and laboratory animals is usually much higher than that in cell cultures, our results are of strong translational significance when conducting experimental studies. On the other hand, the present study has important clinical implications, since several groups of patients, including cancer patients, have impaired albumin levels, which in turn could determine both treatment efficacy and the adverse effects including drug-related toxicity.

## Figures and Tables

**Figure 1 ijms-20-04734-f001:**
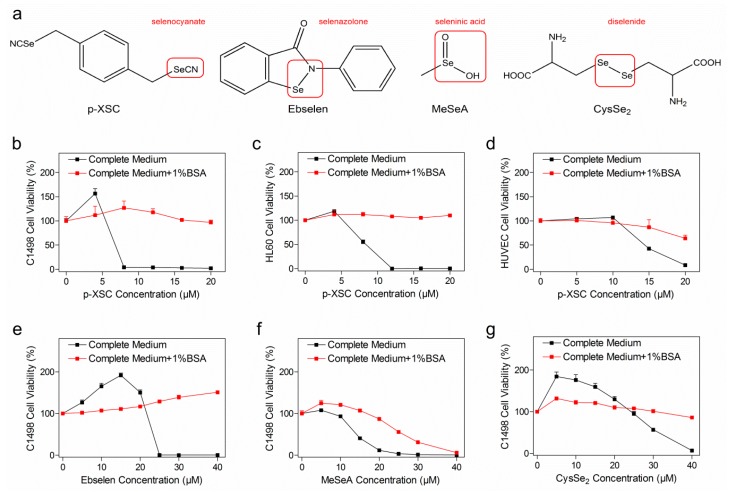
Cytotoxicity of selenocompounds (SeCs). (**a**) Chemical structures of SeCs with the functional groups marked in red. (**b**–**d**) C1498 (murine acute myeloid leukemia), HL60 (acute promyelocytic leukemia), and HUVEC (human umbilical vein endothelial cell) cell viability after p-xyleneselenocyanate (p-XSC) treatment. (**e**–**g**) C1498 cell viability after treatment with ebselen, methylseleninic acid (MeSeA), or selenocystine (CysSe_2_). In all experiments, the exposure time was 24 h and cell viability was assayed by WST-1 kit. The treatment was performed in appropriate complete medium for each cell line without (black line and dot) or with (red line and dot) 1% extra BSA. Results are shown as the mean ± standard deviation of six biological replicates.

**Figure 2 ijms-20-04734-f002:**
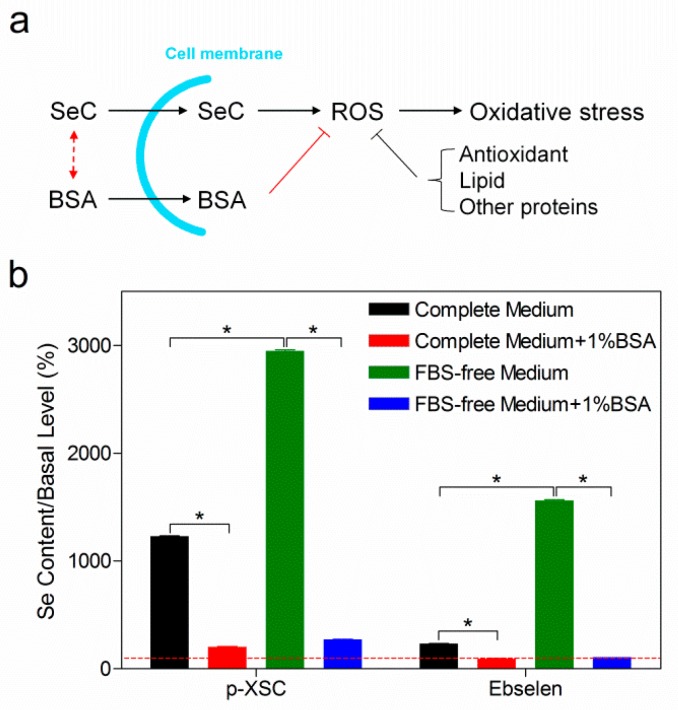
Effect of albumin on the cellular uptake of selenocompounds. (**a**) The SeC enters the cell and generates reactive oxygen species (ROS), inducing oxidative stress and subsequent cell death. BSA might interfere with SeC cytotoxicity through two mechanisms: (1) binding SeC extracellularly (the red dash line with arrows) and modulating the uptake; (2) neutralizing intracellular ROS (the red solid line with end; T-bar) and relieving oxidative stress without affecting uptake. Other cellular components including antioxidant, lipids, and other proteins are also capable of neutralizing ROS (the gray solid line with end; T-bar). (**b**) Intracellular Se level after p-XSC (10 µM) or ebselen (20 µM) treatment for 30 min. The treatments were performed in complete medium (black bar), complete medium supplemented with 1% extra BSA (red bar), FBS-free medium (green bar), or FBS-free medium supplemented with 1% extra BSA (blue bar). Results are shown as the mean ± standard deviation of three technical replicates. The red dashed line marks the basal level from untreated cells. Two-sided Mann–Whitney test was applied to compare the means between groups. * denotes *p* ≦ 0.05.

**Figure 3 ijms-20-04734-f003:**
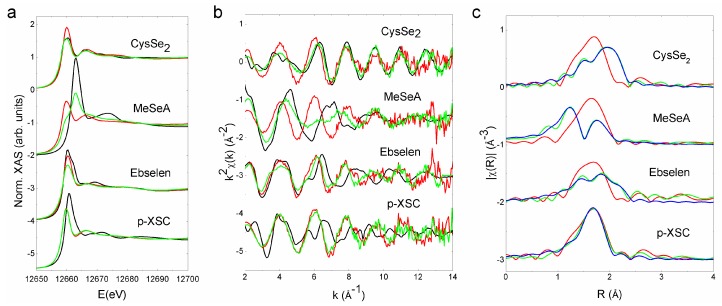
X-ray absorption spectroscopy measurements. (**a**) X-ray absorption near edge spectra of SeC, SeC–BSA, and SeC–FBS. (**b**) Extended X-ray absorption fine structure of SeC, SeC–BSA, and SeC–FBS. (**c**) Pseudo-radial distribution functions of the extended X-ray absorption fine structure for SeC–BSA and SeC–FBS. Experimental spectra of SeC, SeC–BSA, and SeC–FBS are represented as black, red, and green lines, respectively. The blue line in panel (**c**) marks the best fit of the SeC–FBS spectrum.

**Figure 4 ijms-20-04734-f004:**
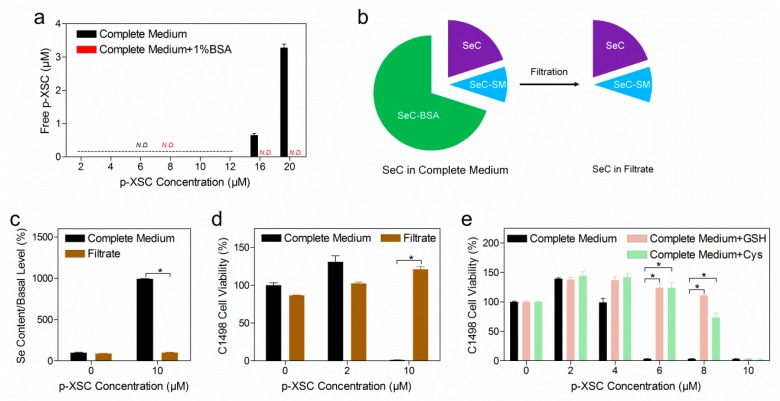
Implication of small-molecule thiol in p-XSC transformation. (**a**) Quantification of free p-XSC in complete DMEM medium without (black bar) or with (red bar) 1% extra BSA. Results are shown as the mean ± standard deviation of three technical replicates. N.D. refers to not detectable. (**b**) Removal of macromolecular SeC–BSA conjugate in complete medium through the filtration method, SeC–SM (conjugate of SeC with small-molecule thiols). (**c**) Intracellular Se level after p-XSC treatment. C1498 cells were treated for 30 min with p-XSC that was provided in form of complete medium or the filtrate. Results are shown as the mean ± standard deviation of three technical replicates. (**d**) C1498 cell viability after p-XSC treatment. p-XSC was provided in form of complete medium or the filtrate. (**e**) C1498 cell viability after concomitant treatment of p-XSC and GSH (10 µM) or Cys (10 µM). In experiments related to panels (**d**,**e**), the exposure time was 24 h and cell viability was assayed by WST-1 kit; results are shown as the mean ± standard deviation of six biological replicates. A two-sided Mann–Whitney test was applied to compare the means between groups. * denotes *p* ≦ 0.05.

**Figure 5 ijms-20-04734-f005:**
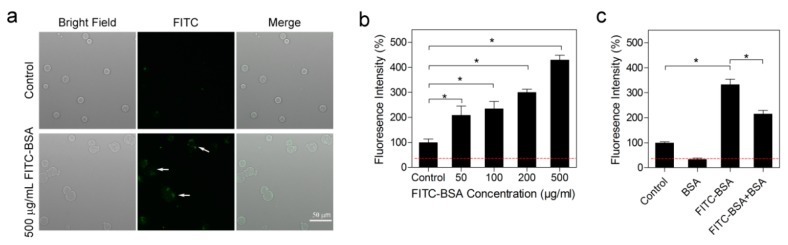
Cellular uptake of fluorescently labeled albumin. (**a**) C1498 cells were treated with FITC–BSA (500 µg/mL) for 14 h and imaged by fluorescence microscope. (**b**) C1498 cells were treated with FITC–BSA (50–500 µg/mL) for 14 h and analyzed for intracellular fluorescence intensity by plate reader. (**c**) C1498 cells were treated with BSA (500 µg/mL), FITC–BSA (500 µg/mL), or FITC–BSA plus BSA (500 µg/mL each) for 14 h and analyzed for intracellular fluorescence intensity. In panel (**b**,**c**), the red dashed line refers to the basal level, and the control group referred to the cells that were briefly mixed with FITC–BSA (500 µg/mL) and immediately rinsed. One-sided Mann–Whitney test was applied to compare the means between groups. Results are shown as the mean ± standard deviation of three biological replicates. * denotes *p* ≦ 0.05.

**Table 1 ijms-20-04734-t001:** Fitting results of the spectra of SeC–FBS considering the Se–S bond in the model. The k-range used was 2.5–13.0 Å^-1^. The amplitude reduction factor (S_0_^2^) was fixed to 0.85.

SeC	Atom	Number	E_0_ (eV)	R (Å)	σ^2^ (Å^2^)	R-Factor
CysSe_2_	C	1	4.8	2.003	0.006	0.022
	S	0.4		2.139	0.004	
	Se	0.6		2.322	0.001	
MeSeA	O	1	4.0	1.682	0.003	0.008
	C	1		2.359	0.004	
	S	1		2.203	0.008	
Ebselen	C	1	5.0	1.920	0.004	0.016
	S	0.4		2.191	0.002	
	Se	0.6		2.355	0.004	
p-XSC	C	1	6.5	2.009	0.003	0.017
	S	1		2.185	0.002	

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
