# Peer review of "Extracellular Albumin Covalently Sequesters Selenocompounds and Determines Cytotoxicity"

_ijms, 2019, doi:10.3390/ijms20194734_

Round 1

Reviewer 1 Report

Authors investigated the effects of BSA on SeCs-induced the toxicity of cancer cells. They report that BSA reduce the uptake and toxicity of p-XSC und Ebselen on cancer cells; however, The inhibitory manners of BSA have not seen on MeSeA und CysSe.  Besides BSA, GSH und Cys can be bound to  p-XSC and thus, cellular uptake of Se should be decreased. In general, i found the manuscript interesting .  The following suggestions are meant to improve the quality:

1) the title can be written ,Extracellular albumin covalently sequesters p- xyleneselenocyanate and determines cytotoxicity .2) fig4. What does Se-SM mean. 3)there is no more mention of how interaction of BSA und p-XSC on the clinical application. 

Author Response

Review 1

Authors investigated the effects of BSA on SeCs-induced the toxicity of cancer cells. They report that BSA reduces the uptake and toxicity of p-XSC und Ebselen on cancer cells; however, the inhibitory manners of BSA have not seen on MeSeA und CysSe.  Besides BSA, GSH und Cys can be bound to p-XSC and thus, cellular uptake of Se should be decreased. In general, I found the manuscript interesting. The following suggestions are meant to improve the quality:

1) the title can be written, Extracellular albumin covalently sequesters p- xyleneselenocyanate and determines cytotoxicity .2) fig4. What does Se-SM mean? 3) There is no more mention of how interaction of BSA and p-XSC on the clinical application. 

Reply: We thank the reviewer for the careful and insightful comment.

1) the title can be written, Extracellular albumin covalently sequesters p- xyleneselenocyanate and determines cytotoxicity

Answer: We agree with the referee that p- xyleneselenocyanate has shown the highest response. However, while assessing the cytotoxicity of MeSeA and CysSe2 after 24 hr treatment, we did found that addition of albumin decreased their efficacy in similar manner as p-XSC and ebselen (Figure 1f and 1g). Moreover, intracellular selenium content after 30 min treatment with MeSeA and CysSe2 was significantly decreased by additional albumin (Figure S2b). However, a short treatment period of 2 hr was not adequate for MeSeA and CysSe2 to fully exert the cytotoxic effect (Figure S1c and S1d). In general, we thought the conclusion regards the pivotal role of extracellular albumin applies to at least four (categories of) selenocompounds, rather than only p- xyleneselenocyanate, and hence we believe that “Extracellular albumin covalently sequesters selenocompounds and determines cytotoxicity” is more appropriate.

2) fig4. What does Se-SM mean?

Answer: SeC-SM stands for the conjugate of SeC with small-molecule thiols. This is added in the revised manuscript Page 8 line 179.

3) There is no more mention of how interaction of BSA and p-XSC on the clinical application.

Answer: We have added the following statements to mention the clinical application of the interaction between BSA and selenocompounds in Page 8 Line 253-258.

“Noteworthy, albumin content in the most used cell culture media is around 0.2% (range 0.1-0.4%) while laboratory animal is 3% and human plasma is 5 %  (ranged 3.5-5.5%). This variability might account for the widely reported heterogeneity in cytotoxicity and alert potential discrepancies in translational studies. Furthermore, albumin level might fluctuate dramatically depending on the pathophysiological background underlying the disease, which can in turn determine the treatment efficacy as well as adverse effects including drug related toxicity of SeCs”.

Reviewer 2 Report

In the present manuscript Zheng and co-workers reported the effect of albumin on cytotoxic potency of selenocompounds. They demonstrated that albumin and thiols bind selenocompounds negatively impacting on drug activity. The manuscript sounds and is well written.  

Author Response

Review 2

In the present manuscript Zheng and co-workers reported the effect of albumin on cytotoxic potency of selenocompounds. They demonstrated that albumin and thiols bind selenocompounds negatively impacting on drug activity. The manuscript sounds and is well written.

Reply: We thank the reviewer for the encouraging comments.